# Driving Accidents, Driving Violations, Symptoms of Attention-Deficit-Hyperactivity (ADHD) and Attentional Network Tasks

**DOI:** 10.3390/ijerph17145238

**Published:** 2020-07-20

**Authors:** Seyed Hojjat Zamani Sani, Zahra Fathirezaie, Homayoun Sadeghi-Bazargani, Georgian Badicu, Safyeh Ebrahimi, Robert Wilhelm Grosz, Dena Sadeghi Bahmani, Serge Brand

**Affiliations:** 1Department of Motor Behavior, Faculty of Physical Education and Sport Science, University of Tabriz, Tabriz 5166616471, Iran; Z.fathirezaie@tabrizu.ac.ir (Z.F.); s_ebrahimi9899@yahoo.com (S.E.); 2Road Traffic Injury Research Center, Tabriz University of Medical Sciences, Tabriz 5166616471, Iran; homayoun.sadeghi@gmail.com; 3Department of Physical Education and Special Motricity, Faculty of Physical Education and Mountain Sports, Transilvania University of Brasov, 500068 Braşov, Romania; wilhelm.grosz@unitbv.ro; 4Center for Affective, Stress and Sleep Disorders (ZASS), Adult Psychiatric Clinics (UPKE), University of Basel, 4002 Basel, Switzerland; dena.sadeghibahmani@upk.ch (D.S.B.); serge.brand@upk.ch (S.B.); 5Substance Use Prevention Research Center and Sleep Disorder Research Center, Kermanshah University of Medical Sciences (KUMS), Kermanshah 67198511115, Iran; 6Departments of Physical Therapy, University of Alabama at Birmingham, Birmingham, AL 35209, USA; 7Department of Sport, Exercise and Health, Division of Sport Science, University of Basel, 4052 Basel, Switzerland; 8School of Medicine, Tehran University of Medical Sciences, Tehran 1416753955, Iran

**Keywords:** driving behavior, attention-deficit/hyperactivity (ADHD), attentional network task, age, traffic accidents, traffic violations

## Abstract

Background: Iran has serious problems with traffic-related injuries and death. A major reason for traffic accidents is cognitive failure due to deficits in attention. In this study, we investigated the associations between traffic violations, traffic accidents, symptoms of attention-deficit/hyperactivity disorder (ADHD), age, and on an attentional network task in a sample of Iranian adults. Methods: A total of 274 participants (mean age: 31.37 years; 80.7% males) completed questionnaires covering demographic information, driving violations, traffic accidents, and symptoms of ADHD. In addition, they underwent an objective attentional network task (ANT), based on Posner’s concept of attentional networks. Results: More frequent traffic violations, correlated with lower age and poorer performance on the attentional network tasks. Higher symptoms of ADHD were associated with more accidents and more traffic violations, but not with the performance of the attentional tasks. Higher ADHD scores, a poorer performance on attentional network tasks, and younger age predicted traffic violations. Only higher symptoms of ADHD predicted more traffic accidents. Conclusions: In a sample of Iranian drivers, self-rated symptoms of ADHD appeared to be associated with traffic violations and accidents, while symptoms of ADHD were unrelated to objectively assessed performance on an attentional network task. Poor attentional network performance was a significant predictor of traffic violations but not of accidents. To increase traffic safety, both symptoms of ADHD and attentional network performance appear to merit particular attention.

## 1. Introduction

Compared to Western European countries and the USA, in Iran, the prevalence rate of traffic-related mortality is high [1,2,3]. Specifically, and as summarized in Abdoli et al. [1], the World Health Organization [4] reported a ratio of 24.1 traffic deaths per 100,000 people per annum in Iran, compared to a ratio of 3.4/100,000 per annum in Switzerland, 4.3/100,000 per annum in Germany, and 11.6/100,000 per annum in the USA. Fortunately, there has been a slight reduction in traffic-related deaths over the last four to six years. However, despite this trend, traffic accidents remain the second largest cause of mortality in Iran [5,6], and the main cause of injuries requiring surgical intervention [7,8,9]. Traffic accidents are most often the result a driver’s poor driving behavior, while the part played by technical malfunctions is negligible [10]. Here, following the Manchester Driving Behavior Questionnaire [11] the following dimensions of poor driving behavior were identified: aggressive violations (e.g., other drivers’ behavior triggers anger), ordinary violations (e.g., not respecting the maximum speed allowed or stop signals; changing lanes without indicating), errors (e.g., driving inappropriately cautiously on highways or on empty roads), and lapses (e.g., forgetting the speed limits in a given area). In this study, the specific cognitive failure of interest includes symptoms of inattention and impulsivity.

Safe driving is the product of multiple cognitive functions, to process diverse multi-sensory inputs and coordinate them with motor-controlled movements. Plausibly, driver distraction and inattention lead to errors, and can cause failures in driving performance; inattention has been identified as one of the main causes of vehicle crashes [12]. With regard to attention, three attentional network tasks are relevant to driving behavior [13], and, based on the human attention network model [14,15,16,17], three distinct functions of attention have been suggested—alerting, orienting, and executive function or conflict resolution [18]. Weaver and colleagues [18] defined alerting as readiness to respond to incoming signals, while orienting concerns the cognitive ability to shift attention from one object or focus to another object or focus, with the aim of selecting new information. Executive function refers to the cognitive ability to grasp conflicting information and to execute an appropriate response. These authors went on to relate these three distinct cognitive processes to three different topographic brain structures (alerting: right frontal and parietal cortex; orienting: areas near the superior parietal lobe, temporal-parietal junction, and superior colliculus; executive control: loops between the prefrontal cortex and anterior cingulate cortex) and to different neurotransmitter systems as proxies for neurophysiological processes. The Attentional Network Test (ANT) is a tool to test both facets of attention and to predict driving behavior [18]. Attentional network tasks and their interactions in driving tasks have been identified in several investigations [13,14,18,19,20,21]. Various aspects of driving attention have been studied, including attention networks [18,22], attention and search conspicuity and visual context [23], visual attention [24], the effect of age and workload on 3D spatial attention in a dual-task driving [25], the influence of salient distractors in relation to the diversion of attention [26], and the effects of perceptual load and driving duration on the mind wandering while driving [27], along with low driving experience associated with less skilled visual scan patterns [28]. However, investigations of attentional network tasks have not been conducted with drivers in Iran. Therefore, the first aim of this study was to test Iranian drivers on the ANT and to examine the associations of facets of attentional network tasks with drivers’ self-reported traffic violations, traffic accidents, and self-reported dimensions of attention-deficit/hyperactivity disorder.

Distractors interfering with attention and perception may also disrupt driving behavior [29,30], and it is possible that such distractors increase the risk of road accidents [31,32]. In one study, for example, factors impairing attention were identified as causes in 905 crash events over a of 36-month period among 3500 drivers [33], and attention errors appeared to be the most common factor in left-turn accidents [34]. On the other hand, higher perceptual skills and a better ability to identify and select relevant stimuli with a shorter reaction time were found to reduce accident among novice drivers [35]. Similarly, higher alertness of drivers to the presence of subjects was associated with higher search conspicuity [23]; furthermore, drivers were quicker to brake and retain steering control when cognitive resources were not depleted by visual distractors [36].

With regard to attention, the neuropsychological condition known as attention-deficit/hyperactivity disorder (ADHD) deserves particular attention. By definition, more severe symptoms of ADHD correlate with poorer decision making [37], lower impulse control and attention, and higher irritability [38]. ADHD is the most common neurobiological disorder in pediatric psychiatry: following Polanczyk and colleagues, the prevalence rates for ADHD are about 5.6%, with no geographical variation in prevalence or incidence over the last 30 years [39,40,41]. With regard to ADHD in adults, prevalence rates of about 4% have been reported [42,43,44]. It is clear therefore that ADHD persists into adulthood [45,46]. Adult ADHD is associated with a higher risk of substance use disorder, and with problems at work and in family life [47], in particular when the symptoms persist and are not treated [48]. On the other hand, from the perspective of evolutionary psychology [49] and evolutionary psychiatry [50], the symptoms of ADHD are also related to advantages in survival and, for example, in creativity [51,52]. With regard to links between ADHD traits and driving behavior, Fuermaier et al. [53] reviewed results from driving simulation studies with adults showing symptoms of ADHD, and found that this group had slower and more variable reaction times, more driving errors, more collisions and crashes, more speeding, and also less control over their poorer steering behavior when compared to adults without ADHD traits. However, very little evidence of this kind has been gathered in Iran. In a previous study, Zamani Sani et al. [22] showed that, for adult drivers in Iran, self-reported symptoms of ADHD were associated with more frequent self-reported traffic violations and traffic accidents. A second aim of the present study was to determine whether these findings could be replicated, but employing different tools (here: ANT; previously [22]: CogLab^®^) to assess different cognitive processes (here: alerting, orienting, executive functions; previously [22]: visual search and spatial cueing).

Previous research [54,55,56] has used the ANT to identify cognitive characteristics associated with ADHD. Several studies have proposed that symptoms of ADHD arise from a primary deficit in a specific executive function domain such as response inhibition or working memory or a more general weakness in executive control [57]. This hypothesis is based on the observation that prefrontal lesions sometimes produce behavioral hyperactivity, distractibility, and impulsivity, as well as deficits in executive function tasks. ADHD is associated with specific limitations in executive function domains [55,56,58]. Symptoms of ADHD appear to be associated with poorer functioning in areas such as visual attention perception and attentional network tasks. Unsurprisingly, ADHD traits correlate with more frequent traffic accidents and riskier driving [59,60,61,62], while cognitive and behavioral driving deficits disappear when individuals with ADHD take appropriate medications, such as methylphenidate [63]. Following Cortese et al.’s systematic review and network meta-analysis [64], methylphenidate is considered the preferred medication for the short-term treatment of ADHD at all ages. In this context, Barkley and Cox [61] noted that the treatment of ADHD traits with appropriate medication correlated with improved driving performance and reduced safety risks in adults with ADHD.

Vaa [65] showed in his meta-analysis that, compared to adults with no ADHD, the risk of traffic accidents was 1.36-fold higher in adults diagnosed with ADHD. Importantly, Vaa [65] included in his meta-analysis only those studies in which participants were thoroughly assessed and diagnosed on the basis of current psychiatric classification systems. Given this, it appears that the long-standing claim of a 4-fold risk of traffic accidents among ADHD-drivers was based on vague diagnoses and data assessed from adolescents and young adults in the late 1980s and early 1990s [66]. ADHD traits have been found, in some studies, to be associated with slower reaction times and poorer behavioral performance in visual-spatial attention [67], and with lower scores for visual attention [68], but not in other studies [69].

With respect to the heterogeneous pattern of results on the association between symptoms of ADHD and driving, methodological issues, the fact that not all individuals with ADHD show more risky driving behavior, and diagnostic issues are all possible contributors. Nonetheless, there are reasons to expect that adults with ADHD are at greater risk of causing motor vehicle accidents than adults without ADHD [70].

As a further contribution to this complex pattern, in a previous study [22], we found that self-rated symptoms of ADHD were associated with self-reported traffic violations, but also with faster objective performance on visual search and special cueing tasks as proxies for cognitive performance.

Three key observations can be made at this point: 1. the associations between symptoms of ADHD and driving behavior appear to be mixed; 2. the moderating effect of symptoms of ADHD on the relationship between attentional network tasks and driving behavior remain unclear; 3. there remains very little research on these issues.

Next, higher age and slower visual processing speed are associated [17,71]. Here, we expected that higher aging would be associated with declining attentional network functioning.

With regard to driving experience, there is evidence that, compared to novices, experienced drivers commit fewer attention-related driving errors [72,73]. Likewise, Zheng et al. [28] showed that experienced drivers had more skilled visual scan patterns than inexperienced drivers. In contrast, it appears that the association between driving experience and symptoms of ADHD is not clear. We therefore introduced driving experience (along with driving frequency) as possible additional factors in the relation between driving accidents and symptoms of ADHD.

Last, it might be expected that, compared to traffic non-offenders, traffic offenders will report more traffic accidents, though the association between traffic-offender status and performance on attentional network tasks and self-reported symptoms of ADHD remains unclear. Given this uncertainty, another aim of the present study was to compare traffic offenders with traffic non-offenders with regard to the symptoms of ADHD, attentional network task, and other aspects of driving behavior.

Based on the evidence summarized above, we formulated three hypotheses and four research questions. Following others [18,23,25,74], we expected poor driving behavior as reflected in traffic violations and accidents to be associated with lower objectively assessed performance on attentional network tasks. Second, following others [22,70], we anticipated that more severe symptoms of ADHD would be associated with poor driving behavior, such as reported traffic violations and accidents. Third, following others [17,71], we predicted a correlation between age and poor functioning of attentional network task. Vaa [65] and Fuermaier et al. [53] both reported a negative correlation between symptoms of ADHD and objective cognitive performance, such as visual search and spatial cueing. However, Zamani Sani et al. [22] were unable to confirm this pattern; in their study, self-reported symptoms of ADHD were associated with better performance on visual search and spatial cueing [22]. Our second research question asked which of the dimensions of age, symptoms of ADHD, and attentional network functioning best predicts traffic accidents and traffic violations. The third research question concerned the strength of the association of driving frequency and driving experience with the functioning of attentional network task and symptoms of ADHD. The fourth research question asked to what extent traffic offenders and traffic non-offenders as defined officially differ with regard to other traffic-related variables, performance on attentional network task, and symptoms of ADHD.

We believe that the present study has the potential to clarify the complex pattern of associations between driving behavior, age, self-reported symptoms of ADHD, and objective cognitive executive processes. The findings may help inform stakeholders concerned with reducing traffic accidents in Iran.

## 2. Materials and Methods

### 2.1. Participants

A total of 274 adults (mean age: M = 31.37, SD = 9.75; 221 males, 53 females) took part in the study. As in the previous study [22], the inclusion criteria were as follows: 1. age between 18 and 65 years; 2. valid driving license; 3. willing and able to follow and to adhere to the study conditions; 4. normal eye-sight, or eye sight corrections providing normal visual acuity. 5. signed written informed consent. Exclusion criteria were: 1. current somatic or psychiatric issues that might negatively influence adherence to the study conditions; 2. current intake of mood, sleep, and alertness-altering substances, such as alcohol, cannabis, opioids, along with sedative medications; 3. dropping out of the study; 4. being left-handed.

### 2.2. Procedure

As in the previous study [22], adults meeting the above-mentioned inclusion and exclusion criteria were asked to participate at this study. Eligible participants were fully informed about the study aims and the anonymous data handling. Thereafter, they signed a written informed consent form and completed a series of questionnaires covering demographic data, driving behavior (see below), and symptoms of ADHD (see below). Next, they underwent an objective measurement of attention and perception at the Motor Behavior Lab of the University of Tabriz (Tabriz, Iran).

Overall, participants needed about 60 min to completing the entire assessment (with 10 min for the cognitive testing). The ethical committee of the Tabriz University of Medical Sciences (Tabriz, Iran) approved the study, which was performed in accordance with the rules laid down in the Declaration of Helsinki and its later amendments.

### 2.3. Tools

#### 2.3.1. Demographic Information

Participants completed a questionnaire on demographic information (age, gender).

#### 2.3.2. Driving-Related Information

As in the previous study [22], participants’ reports related to their preferred vehicle (car; motorcycle), the number of accidents, and the at-fault accidents (those for which the participant was judged to be legally responsible). Next, participants reported on their driving violations; the questionnaire covering this contained 21 items as proposed by the Office of Applied Research of Traffic Police in the Law Enforcement Force of Iran. Typical items were driving too fast, hazardous overtaking, phoning and texting while driving, crossing solid lines, and similar. Answers were given on four-point Likert scales ranging from 1 (= not at all) to 4 (= always), with higher sum scores reflecting a higher frequency of traffic violations. Possible sum scores ranged from 21 (no violations at all) to 84 (several and repeated violations).

We also asked participants to indicate their driving experience (in years) and their frequency of driving (1 = once the month; 2 = once a week; 3 = every day), with higher scores reflecting higher driving frequency.

Next, based on information taken from the official penalty register, participants were labeled as traffic offenders or traffic non-offenders.

#### 2.3.3. Dimensions of Attention-Deficit/Hyperactivity Disorder (ADHD)

As in the previous study [22] participants completed the Adult ADHD Self-Report Scale-V1.1 [75,76], to self-assess dimensions of attention-deficit/hyperactivity disorder spectrum. Typical items are: “How often do you have trouble wrapping up the final details of a project, once the challenging parts have been done?”, or, “How often do you have difficulty waiting your turn in situations when turn taking is required?”; answers were given on a five-point rating scale, with the anchor point 0 (= never) to 4 (= very often); higher sum scores reflect higher self-rated symptoms attention-deficit/hyperactivity disorder.

#### 2.3.4. Attention and Perception

To assess attentional network tasks objectively (alertness/vigilance; orientation/selection; executive function/conflict), the following software was employed: “Attentional Network”; retrievable from: https://github.com/docksteaderluke/CRSD-ANT). All participants performed the cognitive assessment individually at the Motor Behavior Laboratory of the University of Tabriz. The laboratory had an average temperature of 21 °C and sufficient artificial light. Participants were comfortably seated in front of a 17-inch monitor (1024 × 768 resolution; Intel^®^ Iris Plus Graphics 640 1536 MB), controlled by an Apple Mac^®^ computer with a 2.3 GHz Intel^®^ Core i5. The screen was placed on a desk of conventional size and height. Participants were asked to sit upright so as to maintain a distance of 60 cm from the screen. The tasks involved participants pressing keyboard buttons. The attentional network test (ANT) was developed to measure the efficiency of each of the attention networks, namely: 1. alertness/vigilance; 2. orientation/selection; and 3. executive function/conflict [14]. For each of the three corresponding tasks, participants pressed a left key for a left pointing central arrow and a right key for a right pointing central arrow. In addition, for all tasks the fixation point was always in the center of the screen as shown in Figure 1 below.

To assess alerting and orienting the procedure was as follows. Before the target appears, four cue conditions are introduced; in addition, the target can be either above or below the fixation point. The no cue condition is the baseline. The center cue appears at the fixation point with alerting involved only. The double cue condition has two cues at the two possible target locations with alerting involved but not orienting. The spatial cue appears at the target location with both alerting and orienting involved. The difference between no cue condition and double cue condition provides an index of the efficiency of the alerting network. The difference between spatial orienting cue and center cue provides an index of the efficiency of the orienting network.

To assess executive control/conflict, two flanker arrows appear on both left and right side of the central target. The flankers are either congruent (pointing to the same direction as the target) or incongruent (pointing in opposite directions). Therefore, the executive control of attention can be measured by subtracting the mean reaction time (RT) of the congruent condition from the mean RT of the incongruent condition. We use the subtraction of reactions times in the ANT as a measure of the efficiency of the networks [18].

Outcome measures were the number of correct answers, and the percentage of correct and incorrect answers.

The duration of the first fixation was 1600 ms; the maximum allowable response time was 1700 ms, and the duration of the final fixation was set to 3500 ms.

The procedure in the present study consisted of 155 trials (1 block of 31 practice trials and 2 blocks of 62 test trials) and took about 10 min to complete. All participants provided written informed consent. This study was approved by the Tabriz University of Medicine Sciences (IR.TBZMED.REC.1395.1138).

### 2.4. Statistical Analysis

Pearson correlations were computed for associations between age, number of accidents, number of traffic violations, symptoms of ADHD, and processing speed of attentional network tasks, along with driving experience and driving frequency. Next, to predict the number of accidents and traffic violations, age, ADHD symptoms, driving experience and frequency, and processing speed of attentional network tasks were introduced as independent dimensions in two multiple regression analyses. Finally, a series of independent t-tests was performed to investigate differences between traffic offenders and non-offenders. The nominal level of significance was set at alpha < 0.05. All statistical calculations were performed with SPSS^®^ 25.0 (IBM Corporation, Armonk, NY, USA) for Apple Mac^®^.

## 3. Results

### 3.1. Sample Characteristics

Of the 274 participants (mean age = 31.65 years (SD = 9.76; 80.7% males) who took part in the study, 89.7% were car drivers and 10.3% motorcycle drivers.

### 3.2. Traffic Accidents, Traffic Violations, Driving Frequency, Driving Experience, and Objective Processing Speed of Attentional Network Tasks

Table 1 reports the descriptive and correlational statistical indices.

Higher numbers of traffic violations were associated with more accidents and with poorer performance on attentional network tasks. A higher number of accidents was unrelated to performance on attentional network tasks.

Greater driving frequency was associated with longer driving experience but was unrelated to performance on attentional network tasks, age, accidents, at fault accidents, traffic violations, or ADHD scores.

Longer driving experience was associated with greater age, more accidents, fewer traffic violations, more frequent driving, better performance on attentional network tasks. Longer driving experience was unrelated to ADHD scores.

### 3.3. Symptoms of ADHD and Driving Behavior

As shown in Table 1, self-reported symptoms of ADHD were positively correlated with self-reported accidents and traffic violations.

### 3.4. Age and Objectively Assessed Functioning of Attentional Network Tasks

As shown in Table 1, age was unrelated to objective performance on attentional network tasks.

### 3.5. Self-Reported Symptoms of ADHD and Objective Performance of Attentional Network Tasks

Self-reported symptoms of ADHD were uncorrelated with assessed performance on attentional network tasks (overall score and subscales of alerting, orienting, executive functions; see Table 1).

### 3.6. Age, Self-Reported Symptoms of ADHD, Driving Experience and Driving Frequency, and Objective Performance of Attentional Network Tasks as Predictors of Self-reported Traffic Violations and Traffic Accidents

Table 2 provides the statistical indices of the two multiple regression analyses, with self-reported traffic violations and traffic accidents as outcome variables and age, self-reported symptoms of ADHD, and performance on attentional network tasks as predictors. The Durbin-Watson coefficients indicated that independences of residuals were satisfactory. Second, multiple regression models sufficiently explained (R and R^2^) the dependent variables.

Self-reported traffic violations were predicted by younger age, higher self-reported symptoms of ADHD, greater driving frequency, and poorer performance with respect to attentional network task (overall score), while attentional network sub-scores and driving experience were excluded from the equation as these variables did not reach statistical significance.

The number of self-reported accidents were predicted by higher self-reported symptoms of ADHD, lower age, greater driving experience and driving frequency, while performance on attentional network tasks (overall score; sub-scores) were excluded from the equation, as these variables did not reach statistical significance.

### 3.7. Additional Computations; Demographic Information, Driving Behavior, and Cognitive Performance between Traffic Offenders and Traffic Non-Offenders

Table 3 reports the descriptive and inferential statistical indices of demographic information, driving behavior and cognitive performance between officially defined traffic offenders (*n* = 135) and traffic non-offenders (*n* = 128).

Traffic non-offenders did not differ from offenders with regard to age, driving experience, driving frequency, attentional network performance or symptoms of ADHD (trivial to small effect sizes).

## 4. Discussion

We found that, among 274 drivers with an average age around 32, higher numbers of self-reported traffic accidents and traffic violations were associated with objectively assessed poorer performance on attentional network tasks, and with higher self-reported symptoms of ADHD. Age was unrelated to attentional network functioning. Against expectations, a higher number of accidents was unrelated to attentional network performance. Next, more frequent self-reported traffic violations and accidents were predicted by a combination of younger age, self-reported symptoms of ADHD, higher driving frequency and longer driving experience. This latter finding supports the notion of driving as a complex integration of multiple cognitive functions that involve attention and personality traits. Last, officially defined traffic offenders and traffic non-offenders did not differ from each other (trivial to small effect sizes; however, it may be worth emphasizing that actual convictions may not accurately capture true violation rates). We believe that the present findings add to and expand upon the current literature in an important way in showing that (self-reported) symptoms of ADHD impacted neither positively nor negatively on objectively measured attentional network tasks. In contrast, the symptoms of ADHD did appear to increase the risks of self-reported traffic violations and accidents.

Here, we consider the hypotheses and exploratory questions.

Our first hypothesis was that self-reported traffic violations and accidents as reflections of poor driving would be associated with poorer functioning of attentional network tasks, as objectively assessed. The hypothesis was only partially supported. Though higher self-reported traffic violations were associated with poorer functioning of attentional network tasks (overall score and executive function), self-reported accidents were entirely unrelated to this area of cognitive performance. Thus, the present results only partly confirm previous findings [18,23,25]. It is possible that comparing only self-reported traffic violations and accidents with objective cognitive performances is responsible for this mismatch, as it may not to take into consideration other possible confounders. Indeed, as shown in Table 2, results from multiple regression analyses support the notion that additional factors, such as age and symptoms of ADHD, must be taken into consideration.

Our second hypothesis was that self-reported symptoms of ADHD would be associated with poor driving behavior, as reflected in reported traffic violations and accidents, and this was supported. Thus, the present results do replicate what has been observed in previous studies both in other countries [70] and in Iran [22].

Our third hypothesis was that older age would be related to poorer functioning of attentional network tasks as objectively assessed, and this was supported. Accordingly, the present results are consistent with findings reported from previous studies [17,71].

Our first research question concerned associations between self-reported symptoms of ADHD and objectively assessed functioning of attentional network tasks. As shown in Table 1, the correlations were negligible. Given this, we could neither support the notion of an association between symptoms of ADHD and poor cognitive performance [53,65], nor confirm the association between ADHD symptoms and better cognitive performance observed in a previous study [22]. More specifically, and contrary to the results of the previous study [22], we were unable to replicate the association between a higher performance on spatial cueing (considered an equivalent to the dimension of Orientation) and higher self-reported symptoms of ADHD. Unfortunately, we cannot directly compare the data gathered in the two studies, as different samples were assessed, and, in particular, different tools were employed. Given this, we might speculate that either methodological differences, or pure differences in performance, or both were responsible for this lack of overlap in results. In other studies, the symptoms of ADHD have been found to be associated with slower reaction times, poorer behavioral performances on visual-spatial attention [67], and lower scores for visual attention [68]. Furthermore, though it is highly speculative, it is also possible that some participants with more severe ADHD symptoms considered the cognitive tasks to be particularly exciting, while others with these symptoms found the tasks to be particularly boring, so that overall, their respective performances cancelled one another out. In this respect, Roca et al. [77] showed that results on the ANT may vary, as a function of vigilance. Given this, it is possible that some participants with higher symptoms of ADHD performed the ANT in an optimal state of vigilance (cf. Yerkes-Dotson-Rule [78]), while others did not. In the absence of direct evidence for the underlying psychological mechanisms, we suggest that methodological factors, such as sample sizes and forms of assessment of cognitive functions and ADHD, could have been responsible for this mixed and inconsistent pattern of results. However, given that ADHD symptoms were associated with poorer driving behavior (see hypothesis 2, and research question 2), ADHD still deserves particular attention in the context of driving behavior.

Our second research question concerned which of the dimensions of age, symptoms of ADHD, attentional network functioning, driving frequency and driving experience would best predict traffic accidents and traffic violations. As reported in Table 2, self-reported traffic violations were predicted by a combination of younger age, higher driving frequency, higher symptoms of ADHD, and the functioning of attentional network task as objectively assessed. This pattern of results further justifies consideration of several demographic, behavioral and cognitive dimensions concomitantly when investigating traffic violations. However, with regard to accidents, symptoms of ADHD were the only statistically significant predictor. We believe that this latter result justifies taking ADHD into consideration when dealing with traffic accidents.

The third research question concerned the extent of association of driving frequency and driving experience with dimensions of attentional network tasks and symptoms of ADHD. As shown in Table 1, longer driving experience and higher driving frequency were associated with better attentional network performance. To a large extent these results mirror previous results [28,72,73] in that, compared to novices/beginners, experienced drivers perform better on cognitive-attentional tasks. In contrast, neither driving frequency, nor driving experience were associated with ADHD symptoms. The pattern of results suggests that “practice makes perfect” and vice versa. Those participants with longer experience of driving and driving weekly or daily were also those with better attentional network performance.

The fourth and last research question concerned the differences between officially defined traffic offenders and non-offenders with respect to other traffic-related dimensions, the functioning of attentional network tasks, and the symptoms of ADHD. We found no such differences. As shown in Table 3, effect sizes were trivial to small, suggesting, thus, that, among a large sample of drivers, formal status as traffic offender or non-offender has no predictive value.

The novelty of the study should be balanced against the following limitations. First, the voluntary character of the study might have biased the sampling of participants and their adherence to the study conditions. Second, by definition, cross-sectional designs do not allow for causal inferences. Third, data gathered under laboratory conditions such as performance on the attentional network tasks might not reflect real life driving behavior, where cognitive and environmental complexity is both higher and more unpredictable. Fourth, self-rated symptoms of ADHD should have been verified against a thorough psychiatric interview performed by experienced psychiatrists or clinical psychologists. For this reason, we have emphasized throughout that participants were assessed on the basis of self-rated symptoms with ADHD, and we do not claim to have assessed adults on the basis of a thorough diagnosis of ADHD. Given this, future studies in this specific field should also consider a full clinical assessment of ADHD. Fifth, unassessed psychological and physiological traits, such as arousal, alertness, daytime sleepiness, motivation, test anxiety, and increased cortisol concentrations as a proxy for increased psychophysiological arousal, might have distorted two or more dimensions in the same or opposite direction. This holds particularly true, as health-related issues, such as depression, anxiety, and poor sleep, might negatively impact testing driving behavior under laboratory conditions [1,10,79]. Sixth, following Cortese et al. [64], the intake of methylphenidate has a beneficial effect on cognitive performance and safe driving behavior; medication intake should be thoroughly assessed in future studies. Seventh, individuals with ADHD show higher variation of attention within a given time period [80,81], thus, making it desirable to assess individuals with ADHD at different times of the day. Eighth, given that poor sleep is associated with ADHD [82], in future, sleep quality should be introduced as a possible confounder. Ninth, Roca et al. [77] showed that performance on the ANT may vary as a function of vigilance; accordingly, future studies should introduce and assess vigilance as a possible confounder. In this regard, functional brain imaging such as functional near-infrared spectroscopy can assess the frontal lobe function, and provides better estimates of attention and executive function [83]. Tenth, with regard to demographic characteristics, we considered only age and sex, while dimensions such as current employment status, level of education, and socioeconomic status might have provided additional relevant information and further enhanced the pattern of results.

## 5. Conclusions

Among a sample of Iranian drivers, higher self-rated ADHD traits were associated with more frequent accidents and traffic violations, while these traits were entirely unrelated to objectively assessed functioning of attentional network task. The pattern of results suggests more intertwined links among cognitive processes, age, symptoms of ADHD, and traffic violations. The findings are clinically relevant, because they point to a differentiated view of the performance and behavior of drivers with ADHD traits. Policymakers and stakeholders could usefully consider the present results in their efforts to increase traffic safety in Iran.

## Figures and Tables

**Figure 1 ijerph-17-05238-f001:**
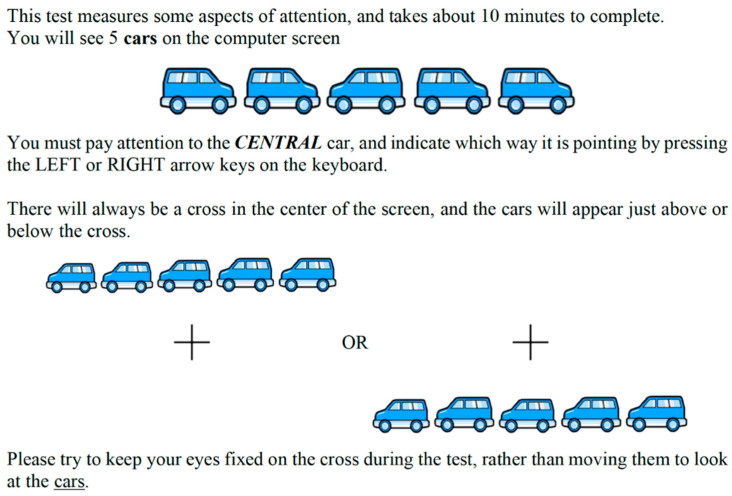
Typical instruction and typical screen scene to solve the cognitive task.

**Table 1 ijerph-17-05238-t001:** Descriptive and correlative statistical indices of demographic driving-related information, symptoms of attention-deficit/hyperactivity disorder (ADHD), attentional network tasks and its functions.

			Dimensions					
Dimensions	Age	Accidents	At-Fault Accidents	Traffic Violations	Driving Frequency ^1^	Driving Experience	ADHD	M (SD)
Age	-	0.06	0.09	−0.32 **	-	-	−0.05	31.37 (9.75)
Accidents	-	-	0.69 **	0.31 **	-	-	0.15 *	1.20 (1.68)
At-fault accidents	-	-	-	0.19 **	-	-	0.16 *	0.53 (1.04)
Traffic violations	-	-	-	-	-	-	0.45 ***	35.28 (9.78)
ADHD	-	-	-	-	-	-	-	1.85 (1.39)
Driving frequency ^1^	0.09	0.09	0.07	0.06	-	-	0.01	1.58 (1.13)
Driving experience	0.82 **	0.18 **	0.12	−0.17 **	0.20 **	-	−0.04	9.30 (7.81)
Attentional network tasks	0.16	−0.06	−0.03	−0.30 **	0.05	0.32 **	−0.07	56.59 (20.13)
Alerting	0.06	0.01	0.01	−0.05	0.08	0.09	−0.10	37.89 (30.56)
Orienting	0.06	−0.04	0.07	−0.04	0.04	0.04	−0.04	45.54 (33.67)
Executive function	0.08	−0.5	−0.11	−0.13 *	0.09	0.12	−0.04	86.35 (38.84)
Correct answers (%)	−0.12	−0.03	−0.04	−0.08	0.04	−0.18	0.03	95.32 (4.67)
Incorrect answers (%)	0.12	0.04	0.03	0.07	−0.04	0.18	−0.03	4.67 (5.25)

Notes: ^1^ Driving frequency: categories: 1 = once a month; 2; = once a week; 3 = every day: ADHD = attention-deficit/**hyperactivity** disorders; * *p* < 0.05, ** *p* < 0.01, *** *p* < 0.001.

**Table 2 ijerph-17-05238-t002:** Multiple linear regressions with traffic violations and number of accidents as dependent variables, and age, symptoms of ADHD, driving frequency, driving experience, attentional network tasks, and their functions as predictors.

Dimension	Variables	Coefficient	Standard Error	Coefficient β	T	*p*	R	R^2^	Durbin-Watson Coefficient
Violations	Intercept	53.96	3.40	-	15.86	0.0001	0.557	0.310	2.119
	Attentional network tasks score	−0.015	0.005	−0.176	−3.031	0.003			
	Symptoms of ADHD	2.164	0.334	0.336	6.479	0.0001			
	Age	−0.223	0.054	−0.242	−4.158	0.0001			
	Driving frequency	3.981	0.754	0.275	5.277	0.0001			
Excluded variables	Driving experience, Alertness, Orientation, Executive function (all t’s < 1.4, all *p*’s > 0.14)								
Accidents	Intercept	2.077	0.505	-	4.114	0.0001	0.319	0.102	1.987
	Symptoms of ADHD	0.176	0.073	0.145	2.421	0.016			
	Age	−0.042	0.018	−0.236	−2.313	0.022			
	Driving experience	0.076	0.023	0.343	3.312	0.001			
	Driving frequency	0.435	0.169	0.159	2.582	0.010			
Excluded variables	Attentional network task score, Alertness, Orientation, Executive function (all t’s < 1.0, all *p*’s > 0.16)								

**Table 3 ijerph-17-05238-t003:** Descriptive and inferential statistical indices of demographic, driving-related information, and attentional network performance between traffic offenders and non-offenders.

	t	Df	*p*	Mean Difference	Std. Error Difference	Cohen’s d	M (SD)	
							Offenders	Non-Offenders
Age	1.058	261	0.291	1.258	1.189	0.13	31.70 (8.6)	30.45 (10.50)
Driving experience	2.748	261	0.006	2.590	0.942	0.34	10.37 (7.28)	7.78 (7.79)
Driving frequency	−2.785	261	0.006	−0.209	0.075	0.34	1.47 (.67)	1.26 (.53)
Attentional network tasks	−0.377	261	0.706	−5.068	13.428	0.05	704.53 (97.87)	709.60 (119.33)
Alerting	1.395	261	0.164	5.313	3.809	0.17	40.57 (31.81)	35.26 (29.84)
Orienting	−1.249	261	0.213	−5.102	4.085	0.15	42.30 (33.04)	47.40 (33.18)
Executive function	−0.640	261	0.523	−3.091	4.828	0.08	85.53 (35.60)	88.63 (42.55)
ADHD	2.879	261	0.004	0.49204	0.17090	0.35	2.09 (1.42)	1.60 (1.34)

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
