# Peer review of "Driving Accidents, Driving Violations, Symptoms of Attention-Deficit-Hyperactivity (ADHD) and Attentional Network Tasks"

_ijerph, 2020, doi:10.3390/ijerph17145238_

Round 1
Reviewer 1 Report
Journal: IJERPH (ISSN 1660-4601)
Manuscript ID: ijerph-804213
Title: Driving accidents, driving violations, symptoms of attention-deficit-hyperactivity (ADHD) and attentional networks
Authors Seyed Hojjat Zamani Sani * , Zahra Fathirezaie , Homayoun Sadeghi- Bazargani , Georgian Badicu , Safyeh Ebrahimi , Wilhelm Robert Grosz , Dena Sadeghi Bahmani , Serge Brand
The work analyses the relationship between:
- Self-reported measures of driving performance (accidents, violations)
- Self-reported data in the ADHD questionnaire. ADHD Self-Report Scale-V1.1 (Green et al., 2018; Jahangard, Haghighi, Bajoghli, Holsboer-Trachsler, & Brand, 2013; Kessler et al., 2005; Ustun et al., 2017)
- Behavioural data measured with the ANT (an attentional task that measures the combined functioning of Posner’s three attentional networks: orientation, executive control and alertness).
It seems to me an interesting study and I believe that it deserves to be published if the suggested changes are followed. In its current form, it seems rather short; I would like to see a more detailed treatment of some of its topics.
I am particularly concerned about these aspects:
Independence from other articles published by the same authors. Have they used the same sample of participants?
“Conclusions: The pattern of results suggests that among a larger sample of Iranian 39 of 274 drivers, self-rated symptoms of ADHD appeared to be associated with higher traffic violations and 40 accidents, while symptoms of ADHD were unrelated to objectively assessed performance on 41 attentional networks”
-Could the Self-reported measure in the ADHD questionnaire compared with the ADHD diagnosis based on clinical interviews?
-The sample seems large (274 people), but the results considered refer to only 39 participants with self-reported symptoms of ADHD. Gender is not specified in this small sub-sample.
It should be made clear whether the authors have recruited different samples for the two articles.
Seyed Hojjat Zamani Sani, Homayoun Sadeghi-Bazargani, Zahra Fathirezaie, Yaser Hadidi, Serge Brand, (2019) Higher symptoms of attention-deficit/hyperactivity disorders (ADHD) and younger age were associated with faster visual perception, but not with lower traffic violations, Transportation Research Part F: Traffic Psychology and Behaviour, 66, 419-429, https://doi.org/10.1016/j.trf.2019.09.010.
In addition, in this recently published article, the authors also tackled the functioning of one of the attentional networks (with a different task). The orienting network is analysed using the Cueing task. It could be discussed whether or not the Orienting results found in the current article (using the ANT task to measure this attentional network) are different to the previous ones reported in the Cueing task (simple, valid and invalid trials) described in the 2019 ms.
The results could also be related to other recent or soon to be published works in which more ecological traffic scenarios are used to measure attentional tasks (specifically orientation).
For instance:
Arexis, M., et al. (2017). Attentional capture in driving displays. British Journal of Psychology, 108(2), 259–275. DOI: 10.1111/bjop.12197
Muela, I et al. (accepted). Visual attention in realistic driving situations: attentional capture and hazard prediction. Applied Ergonomics.
Pierce, R. S., & Andersen, G. J. (2014). The effects of age and workload on 3D spatial attention in dual-task driving. Accident; analysis and prevention, 67, 96–104. https://doi.org/10.1016/j.aap.2014.01.026Posner, M. I. (1980). Orienting of attention. Quarterly Journal of Experimental Psychology. 1980, 32 (1): 3–25. DOI: 10.1080/00335558008248231.
Zheng, X et al. (2020). The effect of leftward bias on visual attention for driving tasks, Transportation Research Part F: Traffic Psychology and Behaviour, 70, 199-207. DOI: 10.1016/j.trf.2020.02.016
I also wonder why the authors think the Attentional Task would behave differently in the Iranian population.
Attentional Network measures (ANT Task) : It would have been logical to analyse the results obtained in the three networks independently rather than jointly. Given that the networks can function in ways opposite to each other, a global measure makes no sense. The abstract does not mention that the behavioural measures are obtained from Posner’s Attentional Measures using the ANT task.
The affirmation established in the abstract should also be discussed, taking into account the previous literature. It seems to me a bizarre result that could be due to an artifact. I don’t know why the authors don’t question its veracity.
“while symptoms of ADHD were unrelated to objectively assessed performance on attentional networks.”
Have a look at some previous works. For instance,
Roca, J.; et al. (2011). Measuring vigilance while assessing the functioning of the three attentional networks: The ANTI-Vigilance task, Journal of Neuroscience Methods, 198(2), 312-324. DOI: 10.1016/j.jneumeth.2011.04.014
The procedure is not well specified. They could have included a more detailed diagram or figure showing graphically how the 3 networks are manipulated.
See examples in the articles I refer to.
Other self-reported measures could be taken.
- They could have related the attentional data with self-reported attentional measures, e.g. the self-reported data in the DBQ (Driver Behaviour Questionnarie): lapses, errors, violations.
-Another important weakness comes from not taking into account socio-demographic measures such as Driving Experience. There is no control measure for the years in possession of a driving license, frequency of driving or the kilometres driven per year. These variables have proved to be relevant factors when considering acquisition of information from the driving environment. For instance, differences have been found between inexperienced, novice and experienced drivers, as well as the moderating effect of experience in driving distraction. A fuller use of sociodemographic data should be made and analyses carried out.
Crundall, D. (2016). Hazard prediction discriminates between novice and experienced drivers. Accident Analysis & Prevention, 86, 47-58. DOI: 10.1016/j.aap.2015.10.006
Horswill, M.S. (2016). Hazard Perception in Driving. Current Directions in Psychological Science, 25(6), 425–430. DOI: 10.1177/0963721416663186
Castro, C. et al. (2019). How are Distractibility and Hazard Prediction in driving related? Role of driving experience as moderating factor. Applied Ergonomics, 81, ISSN:0003-6870 DOI: 10.1016/j.apergo.2019.102886
In addition, the data could be related with other demographic variables, such as recidivism. The sample could be split into offender vs non-offender drivers and the data analysed,
Author Response
See attach,
Thank you!

Reviewer 2 Report
The authors aimed to understand the associations between self-reported symptoms of attention-deficit/hyperactivity disorder (ADHD), retrospective report of traffic violations/accidents, and performance on the Attention Network Test (ANT). This study is important in that both traffic violations and accidents have been shown to be increased in individuals with ADHD. By understanding which attention network(s) is/are associated with driving will help elucidate the neurotransmitters and mechanisms behind increased driving offenses.
While the topic is clinically relevant, there are major areas that require substantial improvements.
1) The concepts of the three attentional skills (alerting, orienting, and executive control) and the experimental variables were poorly described and suggested that the authors required a much deeper understanding of attention and executive functioning.
For example, it would be appropriate for the authors to define words like "alerting", "orienting", and "executive attention". Perhaps this manuscript can describe these three terms using Weaver et al. (2009) in Accident Analysis and Prevention 41, 76-83 as reference:
"Alerting is defined as achieving and maintaining alertness, or readiness to respond to incoming signals; orienting concerns the shifting of attention from one location or object to another in order to select information from sensory input; and executive attention concerns resolution of response conflicts."
At the same time, the authors are encouraged not to use the term "attention network" as an umbrella term for the three facets of attention. The Attention Network Test was originally derived from fMRI studies which described the three attention networks responsible for the different functions of attention. Specifically, alerting attention requires the activation of right frontal cortex and parietal cortex, orienting attention comes from areas near the superior parietal lobe, temporal-parietal junction and superior colliculus, whereas executive control needs the pathways between prefrontal cortex and anterior cingulate cortex. If this study does not study these brain networks through neuroscience tools, it is best to use a different term for the three kinds of attentional skills.
2) I have concerns regarding the difference between this manuscript and the first author's recent publication (citation 13: ZamaniSani et al. Higher symptoms of ADHD and younger age were associated with faster visual perception, but not with lower traffic violations. Traffic Psychology and Behaviour. 2019, 66, 419-29.) The title of the prior paper seems to indicate that the findings of this manuscript are similar; hence, suggesting that there may be overlaps. I personally would have described the differences between the two studies more explicitly instead of stating that "the second aim was to investigate the extent to which previous findings could be replicated" (line 111).
Now I will go section by section to give some constructive comments.
3) Abstract: Sentences that are not directly related to this current study may be removed. If 183 participants completed the study, please describe this group instead of the original 274 participants. As alluded to above, the authors may want to state the actual skills tested in the Attention Network Test, instead of indicating that "the participants’ attentional networks were objectively tested". The results could be summarised based on analysis type. Some of the sentences (1st and 2nd sentences of Results) describe a reverse direction and some sentences (2nd and 4th sentences) seemed to be similar. Perhaps you may want to focus on your regression results (possibly showing coefficients or inferential statistics results) and explain which variables are controlled for.
4) Introduction: This section can be more succinct and focussed on the exact topic being studied. Psychological and psychiatric terms unrelated to this particular study (e.g. evolutionary psychology, survival and creativity, visual attention perceptions, irritability) can be removed in order to avoid confusing readers who are not familiar with these terms. In other words, focus the introduction on defining alerting, orienting, and executive attention, ADHD symptomatology, and the specific driving outcomes captured in this study. It is worth asking a neuropsychologist to read through the Introduction. Lines 112-126 needs a lot of work. Executive function is a higher (top-down) cognitive process, yet attention network is a different entity. Also ADHD is not the same as executive control difficulties. If the goal of the authors is to publish in a non-psychology related journal, it is best to simplify the concepts for readers. Perhaps a discussion on the clinical subtypes of ADHD and how they are associated with driving in prior studies will be beneficial and interesting. Cortese et al's paper has been given a lot of weight in this manuscript, whereas the reviewer knows of many other papers that are equally influential, e.g. MTA and PATS studies, AAP/NICE guidelines, etc.
The paragraph starting with Vaa's study is controversial and seems to be one-sided. Both sides should be described carefully.
Lines 132-137 are simply describing ecological fallacy, which is true for many studies.
5) Methods: It may be helpful to include a figure when explaining the ANT, which is not a simple task (see TIFF file attached). I am not sure if this was a mistake in lines 210-211 "HERE NAME OF THE SOFTWARE". I am still not clear if the authors used the original ANT and whether they used the programme designed by Fan et al. If you did not program your own ANT, please include details of the original ANT (how time is recorded and whether time lapsed until the flanker screen is measured, whether eye trackers are involved, and specify the exact final outcome, e.g. response time, number of errors, or % efficiency). Recruitment strategies should be discussed. It may be helpful to explain why the study included only right-handed individuals. Also for demographic questions, please provide a few samples of questions (e.g. income, education, occupation, etc). Are demographic data considered in the regression models? Allowing participants to pick their preferred vehicle on the questionnaire may introduce bias. Please conduct sensitivity testing and compare data of those who picked to report on a car vs motorcycle. Is the raw score or standard score reported for the Adult ADHD Self-Report Scale (V1)? This will help readers understand the regression models better.
6) Results: Table 1 only showed a heat map and not the exact descriptive statistics (mean/median, SD/interquartile range, etc). It is very important for the authors to include all descriptive data. Inconsistencies are noted, including final sample size, positive vs negative coefficients, and how the subscale scores (Alerting, Orienting, Executive) relates to the composite ANT score. In Table 1, under age, it is rather surprising that the Pearson correlation between age and ANT are so different from alerting, orienting, and executive attention scores. A careful review of the analysis results is warranted.
7) Discussion/conclusion: Description of limitations is well-written. The discussion will improve when #1-6 above are addressed.

Author Response
See attach,
Thank you!

Round 2
Reviewer 1 Report
The manuscript has been significantly improved
and now warrants publication in IJERPH
However, I can not see the supplementary material (i.e. the procedure).
The references are not shown in alphabetic order (including the new ones).
Author Response
Dear Reviewer,
The comments are attached.
Thank you!

Reviewer 2 Report
Dear Authors,
Thank you for revising your previous draft. Please kindly receive my comments for the R1 draft.
1) The term “attentional networks” should be ALL corrected to “attentional network task”, including the Manuscript Title, or to “performance on the ANT” (which the authors had already used in some sentences in the manuscript). There is a difference between a behavioural task of attentional networks and studying attention networks using neuroscience techniques (e.g. PET, MEG, SQUID, ERP+fMRI, neuromodulators, dopaminergic gene, etc). The authors’ one way of examining attention networks through a behavioural task is a good start; however, “attention networks” is not a monolithic concept and can be thought of as more heuristic in nature (see Raz and Buhle, Nature Reviews Neuroscience 7: 367-379, 2006). Some readers may be expecting this paper to describe the effects of the cortical and subcortical attentional network system on driving; hence, being clear from the start that your study has looked at a behavioural task of attentional network is important. For example, in line 34-35, please add the word, “task” to the end of the sentence, “Higher violations, but not accidents, were associated with lower age and a lower performance on the attentional network task.”
2) Thank you for adding the essential information about alerting, orienting, and executive control. This will help readers who may not be familiar with Posner’s theory of attentional networks. The language used to describe attentional network can be simplified somewhat to fit the readership of all medical professionals, including those in public health and environmental health.
3) I read the authors’ reasons for not changing my prior suggestion about the abstract. I still would like you to re-consider your answer. Lines 34-44 are confusing and must be improved, at least in the organisation of it. The content can be the same but without many streams of overlapping findings in separate sentences.
4) Citation #2 is a study talking about car accidents in year 1387. Is this the reference you would like to have? Perhaps if you would like to cite Moradi in your Reference, please kindly use, “An epidemiologic survey of pedestrians passed away in traffic accident”. Journal of Legal Medicine 2003;9(30):75-81. Please be mindful that the first few sentences of the Introduction cannot be justified because the methods used to study road-traffic accidents and mortality are different from that of other countries. Also please provide actual prevalence numbers from countries outside of Iran and also add these references. May be I have missed it, I do not see any reference citing prevalence from other countries. Please kindly read through the epidemiological methods used in each of the studies cited from #1-9 before making general statements that the numbers are higher than other countries.
5) Lines 66-68: This sentence is not scientifically substantiated. I would suggest that the authors remove this sentence. “Symptoms of ADHD” by itself is a well-defined variable and should not be proxies. The authors may consider saying, "In this study, the specific cognitive failure of interest in this study includes symptoms of inattention and impulsivity.” Attentional networks is not a proxy for cognitive information processing. I am concerned that this sentence was in the manuscript
6) Line 83: Please include more than only 3 studies. I believe that one of the reviewers has suggested many more studies related to this.
7) Please kindly note that ADHD is not attention-deficit/hyperarousal disorders. This may be an oversight or a translation error.
8) Kindly note that you do not have sociodemographic information like income, education, etc. Hence please change sociodemographic to "demographic", including the title of Table 1.
9) The following underlined words can be improved either because the tone is somewhat conversational or I suspect minor errors in English grammar:
Line 27-28: Traffic accidents are a dramatic health issue in Iran… (consider changing this to: “A major reason for traffic accidents is cognitive failure due to deficits in attention. In this study, we investigated the associations between traffic violations, traffic accidents, symptoms of attention-deficit/hyperactivity (ADHD), age, and on an attentional network task.”)
Lines 57: “Here, following the Manchester Driving Behavior Questionnaire…”
Lines 59: Is it “red line” or “red light”? I may be wrong because I do not know specifically about the traffic terms in Iran.
Line 63: Perhaps the example given for lapses can be changed. “Forgetting where you have left the car in the car park is not a “poor driving behaviour”. The authors can consider using “unable to maintain alertness” or “inability to filter extraneous distracting stimuli”.
Line 64-65: A careful read of the papers will reveal that “poor general health” and “mental health status with aggression” are also factors. Please distinguish psychological trait vs mental health status.
Line 71-72: “As regards attention”
Line 74: “The ANT as a tool to both test facets of attention and to predict driving behaviour.”
As I suspect that more sentences will require edits (I have not gotten to finish 1 page), I respectfully request the authors to consider English writing supports to assist with sentence formulation and ensure that the authors’ exact ideas can be conveyed appropriately.
Author Response

(The authors gave the same response as above.)
